# Ultimate limit in size and performance of WSe$_2$ vertical diodes

Ghazanfar Nazir[1,2], Hakseong Kim[1], Jihwan Kim[1], Kyoung Soo Kim [3], Dong Hoon Shin[4], Muhammad Farooq Khan[1,2], Dong Su Lee [3], Jun Yeon Hwang[3], Chanyong Hwang[1], Junho Suh[1], Jonghwa Eom[2] & Suyong Jung [1]

Precise doping-profile engineering in van der Waals heterostructures is a key element to promote optimal device performance in various electrical and optical applications with two-dimensional layered materials. Here, we report tungsten diselenide- (WSe$_2$) based pure vertical diodes with atomically defined p-, i- and n-channel regions. Externally modulated p- and n-doped layers are respectively formed on the bottom and the top facets of WSe$_2$ single crystals by direct evaporations of high and low work-function metals platinum and gadolinium, thus forming atomically sharp p–i–n heterojunctions in the homogeneous WSe$_2$ layers. As the number of layers increases, charge transport through the vertical WSe$_2$ p–i–n heterojunctions is characterized by a series of quantum tunneling events; direct tunneling, Fowler–Nordheim tunneling, and Schottky emission tunneling. With optimally selected WSe$_2$ thickness, our vertical heterojunctions show superb diode characteristics of an unprecedentedly high current density and low turn-on voltages while maintaining good current rectification.

[1] Quantum Technology Institute, Korea Research Institute of Standards and Science, Daejeon 34113, Republic of Korea. [2] Department of Physics and Astronomy, Sejong University, Seoul 05006, Republic of Korea. [3] Korea Institute of Science and Technology, Jeonbuk 55324, Republic of Korea. [4] Department of Physics, Ewha Womans University, Seoul 03760, Republic of Korea. These authors contributed equally: Ghazanfar Nazir, Hakseong Kim. Correspondence and requests for materials should be addressed to S.J. (email: syjung@kriss.re.kr)

Unlike graphene, sizable energy band gaps in two-dimensional (2D) semiconducting (SC) transition metal dichalcogenides (TMDs) have drawn enormous attention with the goal to apply SC-TMDs to next-generation electrical and optical devices, including logic circuits, memories, solar cells, light-emitting diodes, and many others[1–13]. The most rudimentary building block in modern electronic and optical devices is one active component in particular: the diode. Efforts in realizing optimal diode operations with 2D van der Waals (vdW) materials have been intensively pursued by fabricating p–n and p–i–n heterojunctions with p- and n-SC-TMDs, as well as insulating (i) 2D layers ever since the mechanical exfoliation of these layered materials was realized[14–16]. It has been found that the performance of 2D vdW-based diodes is heavily influenced by the electric contacts formed at the SC–TMD–metal junctions, either as Schottky or ohmic contacts, and the energy-band alignment in the p–n heterojunctions, either as a type-I or a type-II alignment[17–20]. Therefore, understanding and optimal engineering of the 2D vdW-based diode structures and their operations are essential to meet the full potential that these devices can offer.

Various concepts of 2D vdW-based devices have been proposed, with some demonstrating respectable device performances by utilizing a wide range of SC-TMD energy band gaps varying from a few hundredth meVs to a few eVs, and a straightforward 2D atomic-crystal assembly in lateral or vertical forms[21]. When TMD-based heterojunctions are formed in a vertical stack, devices with these junctions possess several advantages over their counterparts, laterally defined devices[1,14,22]. Vertical p–n junctions, for example, have atomically defined clean interfaces, at which energy band alignments can be precisely tuned for various applications, not to mention that the active channel length of the devices can be readily chosen by the layer number and even scaled down to one-atomic thickness[4,17,18,23–27]. Owing to the crystalline nature of the 2D vdW materials, moreover, the ability to engineer the twist angles defined in vertical heterostructures has added other exciting aspects in device applications such as resonant tunneling diodes[28], tunneling field-effect transistors[29–31], and even superconducting devices[32].

Most experimentally demonstrated vertical devices, however, have several unresolved issues in realizing a true vertical geometry and understanding the exact transport behavior across those atomically sharp junctions[1,14,33]. Numerous reports have claimed that the atomically defined p–n junctions in their vertical heterostructures could be essential for the reported diode characteristics; these claims though are now challenged by recent evidence that the Schottky barriers at metal–SC–TMD interfaces and unaccounted lateral transport channels could induce such strong diode-like charge transport characteristics[14]. In addition, large resistances built at the contacts between metal and SC-TMDs and at the vdW interfaces inside the SC-TMD heterostructures become major obstacles for achieving a high value operating on-current. When fabricating vertical heterostructures, moreover, random stacking and uncontrollable geometrical defects such as bubbles become imminent problems for TMD-based electronics to be employed in real industrial applications[34]. Recently, an atomically clean vertical p–n junction was demonstrated in homogeneous molybdenum disulfide (MoS₂) films by selective chemical doping to relieve the above-discussed electrical and physical issues in the SC-TMD vdW heterostructure, but the on-current level was still limited by the metal–MoS₂ contacts and lower charge–carrier mobility across laterally formed transport channels[33,35].

Here we present a simple but optimal approach to form a true vertical diode with a homogeneous WSe₂ crystal. We find out that the Fermi level ($E_F$) of the WSe₂ films can be efficiently tuned either toward a conduction (n-doping) or a valence (p-doping) band by metal-assisted doping in forming gadolinium (Gd, n-doping) or platinum (Pt, p-doping) contacts to the WSe₂. Since both p- and n-doped regions are confined to the first few layers on each side of

the flakes[35], the ultimate limit in device performance of the vertical diodes is exclusively ruled by the choice of WSe₂-film thickness, or in other words, the width of the un-doped (insulating) region. As the thickness increases from monolayer to multilayers, charge transport through the vertical p–i–n junctions is regulated by a series of quantum tunneling events: direct tunneling (DT), Fowler–Nordheim (FN) tunneling, and Schottky-emission (SE) tunneling. For much thicker films ($t_{WSe_2} \geq 35$ nm), space-charge effects limit the vertical charge flows. Our WSe₂ vertical devices reveal quite intriguing diode characteristics, combining the advantages from both conventional Schottky and p–i–n diodes possess. The WSe₂ vertical p–i–n junctions formed with flakes of $t_{WSe_2} \approx 10$ nm, where FN and SE tunneling events are dominant, reveal superb diode characteristics of an unprecedentedly high on-current (current density) value of $I_{ON} \geq 10$ mA ($J \geq 10^5$ A cm$^{-2}$), even at a modestly low source–drain voltage ($V_{SD}$) of $V_{SD} = 1$ V at $T = 300$ K. Furthermore, because charge transport through our vertical diodes is determined by a series of tunneling effects, the on-current increases by more than six orders of magnitude, following the ideal Shockley diode relation with a diode ideality factor of $n = 1$. In addition, the asymmetric nonlinear $I$–$V_{SD}$ characteristics for forward ($I_{FW}$, $V_{SD} > 0$ V) and backward ($I_{BW}$, $V_{SD} < 0$ V) bias guarantee a reasonably high rectification ratio (RR = $I_{FW}/I_{BW}$) of RR $\geq$ 100 at $T = 300$ K for the homogeneous vertical diodes with $t_{WSe_2} \approx 10$ nm. The rectification behaviors are further improved up to RR $\geq$ 10,000 for the diodes with thicker films ($t_{WSe_2} \approx 42$ nm) with only a modest setback of the on-current values. We ascribe these superb diode characteristics to the formation of ideal p–i–n heterojunctions free of crystallographic misalignments, ultra-low metal–WSe₂ contact resistances, and staggered energy-band alignments inside the vertical channel due to the limited charge transfer between 2D vdW layers. Finally, a high-efficient switch operation in a radio-frequency (RF) domain is demonstrated as a testbed for practical device applications.

## Results

**Fabrication of pure vertical WSe₂ diodes.** Figure 1a, b illustrate 3D diagrams for the WSe₂ vertical diodes fabricated on a silicon nitride (SiN) membrane (Fig. 1a) with a corresponding cross-sectional image (Fig. 1b), depicting the metal-induced doped and intrinsic WSe₂ layers. The SiN membrane with a 3-μm-diameter through-hole (Fig. 1e) is used as the vertical device platform, on which WSe₂ flakes with various thicknesses ranging from monolayer to multilayers are transferred with a micro-manipulation stage. We confirm the layer numbers of WSe₂ flakes with optical contrast (Supplementary Figure 3) and atomic force microscope measurements (Supplementary Figure 4), and thicknesses for some devices are later confirmed by high-resolution transmission electron microscopy (TEM) images (Supplementary Figures 4 and 5). To form pure vertical WSe₂ p–i–n heterojunctions, more than 20-nm-thick Gd and 10-nm-thick Pt films are separately deposited onto the top-most and bottom-most facets of the flakes. Both Gd and Pt are protected by sequentially deposited gold films, prior to any further characterizations. The frustum of the pyramid on the backside of the Si substrate is filled with sputtered titanium (Ti, ≈20 nm) and gold (Au, ≥400 nm) films to ensure electrical connections. We choose Gd ($\Phi_{Gd} \approx 2.9$ eV) and Pt ($\Phi_{Pt} \approx 5.6$ eV) as the contact materials to maximize the work–function ($\Phi$) difference in order to facilitate the electrically induced electron and hole dopings on each side of the WSe₂ ($\phi_{WSe_2} \approx 4.3$ eV)[36–38]. Intrinsic WSe₂ flakes are ambipolar semiconductors with $E_F$ aligned in the middle of the energy gap[20,39], which makes it possible to tune $E_F$ either toward a conduction (n-doping, Gd) or a valence (p-doping, Pt) band by the metal-induced dopings. Note that the accumulated charges from the built-in electric fields are confined to an approximately 1-nm-thick band from the junctions (Fig. 1b), so that p- and n-doped regions are expected to be fixed to the first and the

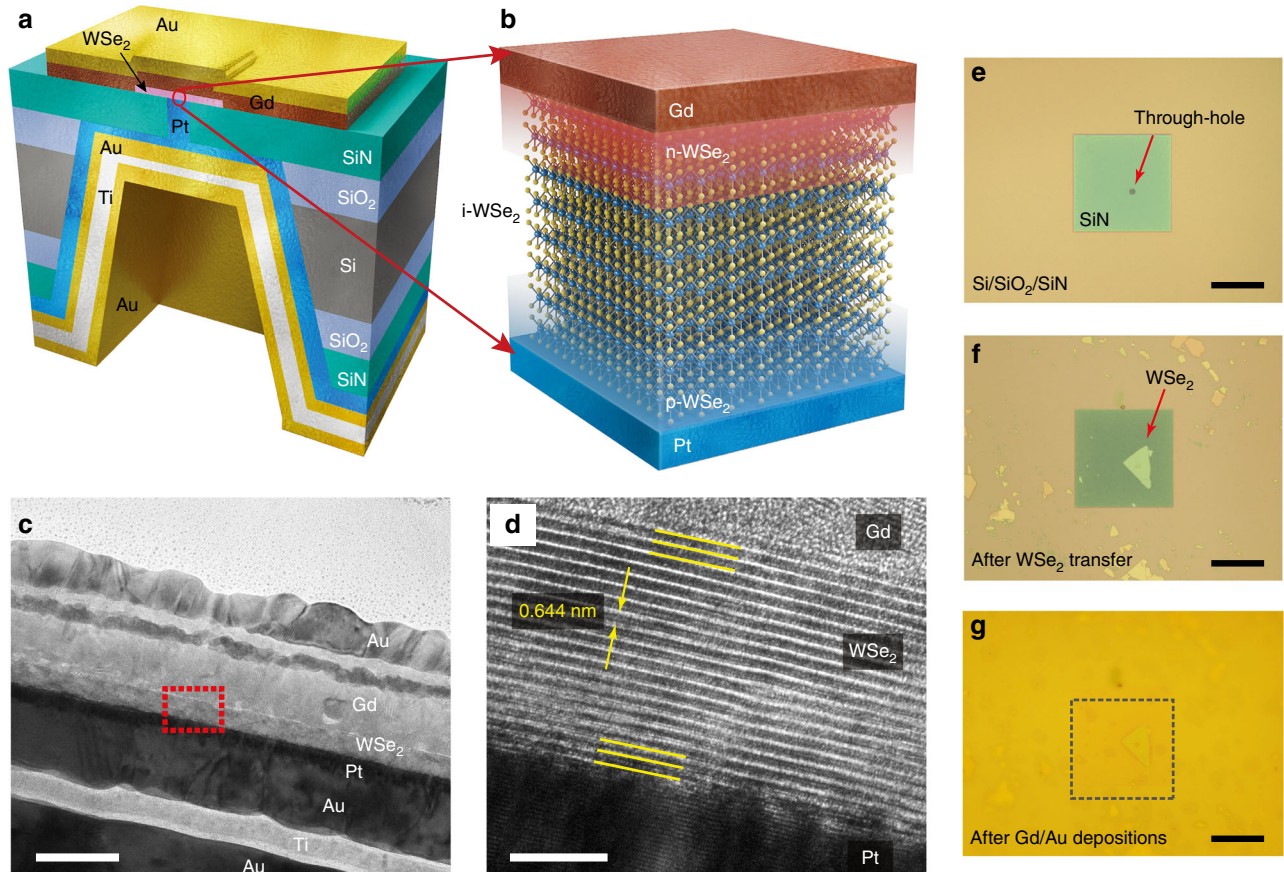

**Fig. 1** Schematics and viewgraphs of vertical WSe$_2$ diodes. **a** Schematic of vertical WSe$_2$ diodes fabricated on a SiN membrane with a through-hole of 3 μm in diameter. True vertical p–i–n heterojunctions in homogeneous 2D vdW materials are formed through the depositions of Gd and Pt films on each facet of WSe$_2$ flakes. Oxidation-prone Gd and Pt films are protected by Au films, and sputtered Ti and Au metals fill the void on the backside of the Si/SiN platform to secure electric connections for later device characterizations. **b** Three-dimensional atomic scale illustration of WSe$_2$ p–i–n heterojunctions. Metal-induced doping is confined within the first few layers of WSe$_2$ flakes from the Gd (Pt)–WSe$_2$ interface, leaving the middle of the WSe$_2$ flake unperturbed (i-WSe$_2$). **c**, **d** High-resolution TEM images displaying the complete set of materials used for fabricating the vertical WSe$_2$ diodes (**c**, scale bar 50 nm), and the layered structure of WSe$_2$ crystals (**d**, scale bar 5 nm). **e–g** Sequential optical viewgraphs taken after fabricating a through-hole on a SiN membrane (**e**), transferring a WSe$_2$ flake on top of the hole by dry-transfer technique (**f**), and evaporating Gd and Au films on top of the flake (**g**). Scale bar 30 μm

second layers of each side of the WSe$_2$ regardless of its thickness, thereby sandwiching the un-doped layers (i-WSe$_2$) to form true vertical homogeneous p–i–n heterojunctions[35]. Within our operation voltage range ($|V_{SD}| \leq 1$ V) in the vertical diodes, the intrinsic WSe$_2$ can be considered as an insulating layer. Here we note that the intrinsic WSe$_2$ layers in our vertical diodes are quite unique when compared with the insulating area defined by the depletion region in conventional bulk SC p–n heterojunctions. While the width of the depletion region (i.e., the size of the insulating region) is tuned by the polarity and magnitude of $V_{SD}$ in regular p–n diodes, the i-WSe$_2$ layers are less dependent on varying $V_{SD}$ owing to weaker interlayer interactions across the vdW gap in 2D-layered vdW materials. Figure 1c shows a TEM image displaying the complete set of materials used in fabricating the vertical WSe$_2$ p–i–n heterojunctions. The high-resolution TEM image (Fig. 1d) reveals that the layered structures of the film are intact even at the interfaces of Gd and Pt films with 0.644 nm as the unit thickness (Supplementary Figure 5), supporting that the metal-induced dopings in our devices are mostly due to the built-in electric fields from the work–function differences at the junctions of Gd–WSe$_2$ and WSe$_2$–Pt.

**Device characteristics of vertical WSe$_2$ diodes.** Figure 2a displays a typical current–voltage ($I$–$V_{SD}$) characteristic curve from the vertical p–i–n heterojunction made of a 10.4-nm-thick, or

16-layer (16L), WSe$_2$ flake at $T = 300$ K. Forward (backward) bias is defined when positive $V_{SD}$ is applied to the bottom Pt (top Gd) contact. The most noticeable feature in our vertical diodes is that they can operate with an unprecedentedly high on-current setting. Even at a modestly low $V_{SD} = 1$ V, the on-current increases up to $I_{ON} > 10$ mA, which equates to a current density of $J \geq 2 \times 10^5$ A/cm$^{-2}$ at room temperature. This extremely high current density, which has yet to be reported from any other nanoscale diodes, not to mention those with 2D vdW materials, assures that our vertical WSe$_2$ devices can be used in high current diode applications. Moreover, the on-currents in our vertical diodes increase by more than six orders of magnitude at a much faster rate than the values for previously reported 2D vdW-based diodes, which results in lower turn-on voltages. The red line in the inset of Fig. 2a is from the numerical fitting to the Shockley diode relation of $I = I_s(\exp(-qV_{SD}/nk_bT) - 1)$, where $I_s$ is the reverse-bias saturation current, $k_b$ is the Boltzmann constant, and $q$ is an elementary charge (see Supplementary Note 1 and Supplementary Figure 11). We set the diode ideality factor to $n = 1$ for the fitting, and the agreement with the data (gray circles) indicates that the vertical WSe$_2$ p–i–n heterojunctions follow the ideal Shockley diode operations at lower $V_{SD}$, while on-currents undergo a six-order increase in magnitude. At last, the vertical diode with a 10.4-nm-thick film reveals a reasonable rectification behavior of RR ≈ 100, maximized at $V_{SD} \approx 0.4$ V.

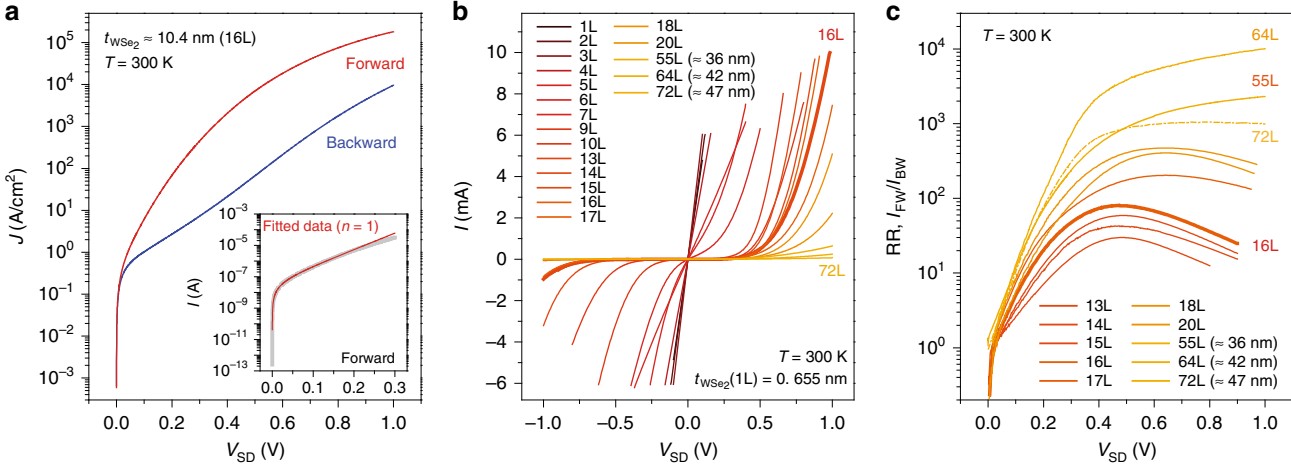

**Fig. 2** Device characteristics of vertical WSe$_2$ diodes. **a** $J$ vs. $V_{SD}$ curves from the device with a 16L WSe$_2$ flake ($t_{WSe_2} \approx 10.4$ nm), revealing a high current density of $J > 2 \times 10^5$ A/cm$^2$ and a reasonable rectification ratio of RR $\approx 100$ at $T = 300$ K. (inset) Log($I$) vs. $V_{SD}$ curve in the forward-biased direction at $T = 300$ K. Current increases by more than six orders of magnitude up to $I \le 10^{-6}$ A (gray circles), following an ideal Shockley diode relation with a diode ideality factor of $n = 1$ (red line). **b** $I$ vs. $V_{SD}$ characteristic curves from the vertical WSe$_2$ heterojunctions with varying WSe$_2$ thickness from a monolayer (1L) to a thick ($\approx 47$ nm) film. At $T = 300$ K, a linear response of the current at an increasing $V_{SD}$ is observed up to a 5L device, and optimal current rectifications are observed from the devices with 13L and thicker films. **c** Rectification ratio (RR, $I_{FW}/I_{BW}$) vs. $V_{SD}$ at different WSe$_2$ thicknesses extracted from the $I$ vs. $V_{SD}$ curves at $T = 300$ K. RR increases in value as the thickness increases because of the enhanced backward-current suppression, and the $V_{SD}$ where RR reaches its maximum value is dependent on the layer number

Device characteristics of our vertical WSe$_2$ diodes are determined by a series of quantum tunneling events, which are sensitive to the thickness of the intrinsic region of the heterojunctions. Figure 2b displays a series of $I$–$V_{SD}$ characteristic curves at $T = 300$ K from the devices with a monolayer to those with much thicker films ($t_{WSe_2} \le 47$ nm). Up to five layers, $I$–$V_{SD}$ curves maintain their linear behavior to $I_{ON} \approx 10$ mA at $T = 300$ K (Supplementary Figure 6). At $T = 6$ K, however, $I$–$V_{SD}$ curves become nonlinear with a moderate current suppression at $V_{SD} \approx 0$ mV for 3L and thicker films (Supplementary Figure 7), implying that the vertical p–i–n heterojunctions composed of 3L and thicker flakes are not purely ohmic junctions. The weak temperature dependence in the $I$–$V_{SD}$ curves suggests that the vertical junctions made of the 1L and 2L flakes are not metallic contacts either. Note that the $I$–$V_{SD}$ curves at $T = 300$ K from the devices with 1L, 2L, and 3L flakes are indistinguishable, mainly due to the fact that the junction resistances of these heavily doped layers become comparable to or even smaller than the combined contact resistance ($\le 15\,\Omega$) of the Gd–WSe$_2$ and WSe$_2$–Pt junctions. We point out that here, in our vertical junctions, we cannot find any charge transport characteristics that could be related to plausible metal-induced structural defects or metal-atom diffusions through the SC films[40], mostly due to the confined area of interest of our WSe$_2$ vertical junctions (3 μm in diameter) and the carefully controlled Pt and Gd metal evaporation conditions.

As WSe$_2$ thickness increases up to $\approx 5.9$ nm (9L), the nonlinear characteristics of the $I$–$V_{SD}$ curves become symmetrically reinforced at both forward and backward $V_{SD}$, which is not an ideal trait for diode operations. When the flakes are in the range of 8.5 nm (13L) $\le t_{WSe_2} \le 13.0$ nm (20L) in thickness, however, current in the backward $V_{SD}$ becomes notably suppressed, while the on-current in the forward bias maintains its value up to $I_{FW} \ge 1$ mA at $V_{SD} = 1$ V, demonstrating respectable diode operations from the pure vertical heterojunctions. Accordingly, RR, which is close to unity for thinner WSe$_2$ flakes ($t_{WSe_2} \le 5$ nm, 7L), increases as high as RR $\ge 10,000$ in value as the WSe$_2$ becomes thicker (Fig. 2c). Moreover, the $V_{SD}$ where RR reaches a maximum value increases as the film becomes thicker, suggesting that our vertical

diodes possess a knack for wide-ranging diode applications, following their characteristic that the operating $V_{SD}$ for efficient rectification can be readily determined by the mere choice of WSe$_2$ thickness.

It is quite interesting to note that our vertical WSe$_2$ diodes possess all the positive features that both conventional 3D semiconductor-based Schottky and p–i–n diodes can provide. For example, Schottky diodes are known for their high value on-current (current density) and low turn-on voltage, characterized with a diode ideality factor close to the ideal case of $n \approx 1$; in general, however, conventional Schottky diodes suffer from weak rectification characteristics. On the other hand, p–i–n diodes have advantages in strong rectification behaviors, but suffer from low operating current and high turn-on voltage. Moreover, beyond our straightforward and easy device fabrication process, our vertical WSe$_2$ diodes possess an additional feature in having the layer number control knob, which is lacking in previously reported 3D and 2D vdW-based diodes.

**Charge transport mechanisms through vertical WSe$_2$ p–i–n heterojunctions.** Figure 3a–c illustrate how energy bands are possibly aligned in the vertical WSe$_2$ junctions with Gd- and Pt-film-induced dopings. As discussed in other reports and suggested in our measurements, the metal-doped regions can be extended to the first few layers, from both Gd–WSe$_2$ and WSe$_2$–Pt interfaces[35]. Therefore, as depicted in Fig. 3a, there exists more than one unperturbed WSe$_2$ layer sandwiched by metal-doped regions for the devices with thicker films, forming a p–i–n heterojunction with Schottky-type tunnel barriers at each contact. Note that the staggered type-II energy band alignment inside the vertical channels is due to the limited charge transfer between the 2D vdW layers through physical vdW gaps. When vertical p–i–n heterojunctions are properly biased in the forward direction as shown in Fig. 3b, a series of DT, FN, and SE quantum tunneling events play dominant roles for both electrons and holes to tunnel either through the junction (DT, FN) or to the energy bands (SE) of the insulating layers. In comparison, when the WSe$_2$ heterojunctions are backwardly biased (Fig. 3c), transport

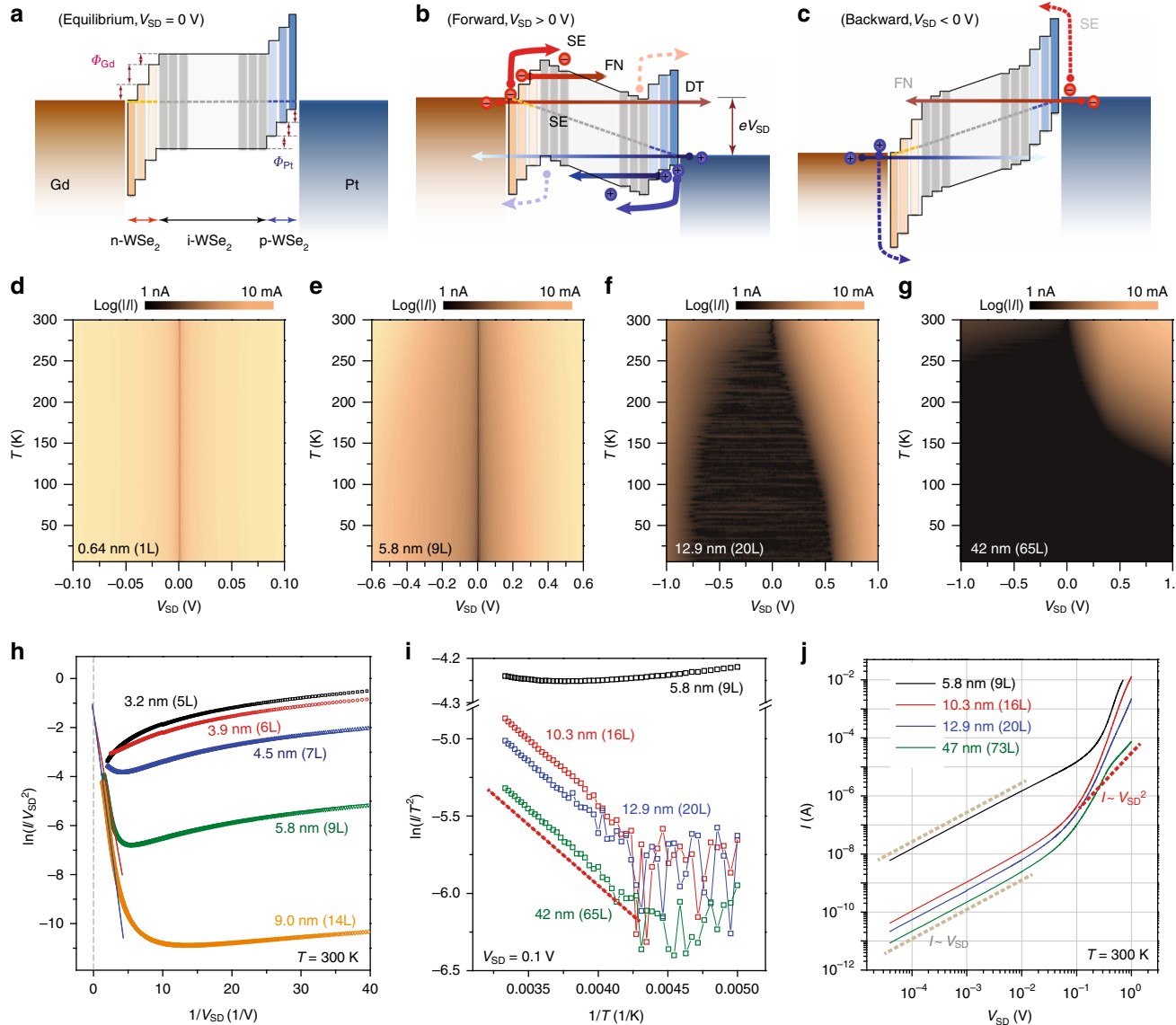

**Fig. 3** Charge transport mechanisms through vertical WSe$_2$ p–i–n heterojunctions. **a–c** Energy band diagrams of the vertical WSe$_2$ p–i–n heterojunctions at equilibrium (**a**), forward (**b**), and backward (**c**) $V_{SD}$. A series of quantum tunneling events through the junctions are denoted with direct tunneling (DT), Fowler–Nordheim tunneling (FN), and Schottky emission tunneling (SE) for both electrons and holes. **d–g** Two-dimensional display of $I$ vs. $V_{SD}$ curves at varying temperatures from $T = 6$ to 300 K for the devices with WSe$_2$ films of 0.64 nm (1L) (**d**), 5.8 nm (9L) (**e**), 12.9 nm (20L) (**f**), and 42 nm (65L) (**g**). **h** $\ln(I/V_{SD}^2)$ vs. $1/V_{SD}$ plots, reconfigured from $I$–$V_{SD}$ characteristics at $T = 300$ K for different WSe$_2$ thicknesses. The distinct linear response in the plots of the 5.8-nm-thick (green) and 9.0-nm-thick (orange) films indicates that FN tunneling is the leading quantum transport mechanism through the junctions. No FN tunneling features are observed from the junctions with 3.9 nm (6L) and thinner films. **i** $\ln(I/T^2)$ vs. $1/T$ relations at $V_{SD} = 0.1$ V from the vertical junctions with varying WSe$_2$ thickness. The thermally activated tunneling events are prevalent in the devices with 10.3 nm and thicker flakes. **j** $I$ vs. $V_{SD}$ curves for different devices plotted in log–log scale. At low $V_{SD}$, current is proportionally increasing to the bias voltage $I \sim V_{SD}$ for all the devices. For a thick ($t_{WSe_2} \approx 47$ nm) film, current is proportional to the square of $V_{SD}$ ($I \sim V_{SD}^2$), a general trait for space-charge-limited transport through low-mobility semiconducting films

through the junctions exclusively relies on DT and FN tunneling events, with suppressed SE effects from the heightened Schottky tunnel barriers.

To understand the transport mechanisms across the vertical WSe$_2$ heterojunctions in depth, we investigate how $I$–$V_{SD}$ characteristics evolve as temperature varies. Figure 3d–g show a series of temperature-dependent measurements for the vertical devices with WSe$_2$ films of thickness 0.64 nm (1L, Fig. 3d), 5.8 nm (9L, Fig. 3e), 12.9 nm (20L, Fig. 3f), and 42 nm (65L, Fig. 3g), from $T = 6$ to 300 K with a spacing of $\Delta T = 2$ K (see Supplementary Figure 8 for other devices). For the device with monolayer film (Fig. 3d), the weak current modulation at varying temperatures and linear $I$–$V_{SD}$ behaviors even at lower

temperature suggest an ohmic contact, although further experimental and theoretical works are required to quantify how many electron and hole carriers are induced from the metal contacts and how the electronic structures of the WSe$_2$ film are modified at the interface between the TMD layer and metal films. As indicated previously, a moderate current suppression at $V_{SD} \approx 0$ mV at low temperatures suggests that there exist tunnel barriers at the junctions of the devices with 3L and thicker films. The weak temperature dependences in $I$–$V_{SD}$ curves suggests that DT contributes heavily to charge transport in the WSe$_2$ p–i–n heterojunctions made of films up to $t_{WSe_2} \leq 3.9$ nm (6L).

When WSe$_2$ films become thicker than 6L and $V_{SD}$ increases, which further modify the tunnel barrier to a trapezoid and then a

triangular shape, FN tunneling across the junctions starts contributing. To confirm FN tunneling, we replot the $I$–$V_{SD}$ responses to a FN relation: $\ln(I/V_{SD}^2)$ vs. $1/V_{SD}$ (Fig. 3h), for the devices with varying layer numbers. The FN tunneling, identified with a linear dependence in $\ln(I/V_{SD}^2)$ vs. $1/V_{SD}$ as guided with solid red and blue lines in Fig. 3h, becomes effective for the 5.8 nm (9L) and 9.0 nm (14L) devices, while it weakly activates for the device with a 4.5 nm (7L) flake and is completely absent for the devices with thinner films. The relatively weak temperature dependence further proves that vertical transport through the p–i–n heterojunction made of the 9L-WSe₂ film is governed by DT at lower $V_{SD}$ and FN at higher $V_{SD}$ (Fig. 3e).

Once the vertical p–i–n heterojunctions are made of 6.5 nm (10L) and thicker films, thermal activations of the charged carriers across the Schottky barriers or thermionic field emissions (Fig. 3b) make $I$–$V_{SD}$ curves temperature dependent, as displayed in Fig. 3f, g. Figure 3i displays the typical Richardson plots of $\ln(I/T^2)$ vs.$1/T$ for the devices with different thicknesses at a forward bias of $V_{SD} = 0.1$ V. Linear relations in the plots, as guided with a dotted red line, prove that thermally activated charged carriers contribute to vertical transport through the devices with thicker films ($t_{WSe_2} \geq 6.5$ nm). In comparison, no thermal activation effects are observed in the device with the 5.8-nm- (9L) thick film (Supplementary Figure 9). We can estimate the effective Schottky barrier height ($\Phi_{SB}$) of the vertical p–i–n heterojunctions from the slope of the linear relations in $\ln(I/T^2)$ vs. $1/T$. In general, $\Phi_{SB}$ extracted from the current variations measured at a higher $V_{SD}$ has a tendency to be underestimated by the contributions of additional conducting channels, such as FN and thermionic field emissions, and short-channel effects[41]. Thus, we extrapolate the effective barrier heights at different $V_{SD}$ to the $\Phi_{SB}$ at the equilibrium condition at $V_{SD} = 0$ V (Fig. 3a), and find out that the $\Phi_{SB}$ in our vertical WSe₂ p–i–n junctions made of Gd and Pt contacts is around $\Phi_{SB} \approx 420$ meV (Supplementary Figure 10).

When the vertical WSe₂ heterojunctions are formed with a much thicker flake ($t_{WSe_2} \geq 35$ nm), the on-current of the vertical diodes is limited by the effects related to the properties of bulk WSe₂, such as a low charge–carrier mobility reducing its current down to sub-mA ranges even at $V_{SD} > 1$ V (Figs. 2b, 3g). Figure 3j shows $I$–$V_{SD}$ curves plotted in log–log scale from different devices at $T = 300$ K. As guided with dotted light-brown lines, charge transport through the junctions is mainly governed by DT at low $V_{SD}$ ($I \sim V$), and subsequently followed by FN and SE tunneling events. Quite noticeably, however, for the device with $t_{WSe_2} = 47$ nm (73L) (solid green line), current increases at a rate of $I \sim V^2$ at $V_{SD} \approx 1$ V, which is an archetypal signature for the space-charge-limited currents commonly observed in low-mobility SC films[42,43].

**Vertical heterojunctions with symmetric vs. asymmetric metal contacts.** In our vertical diodes, asymmetric metal contacts to the WSe₂ flakes are essential to maximize the operating current in forward bias while minimizing the off-current in backward bias, thus promoting current rectification. In order to specify the roles of each component in the vertical diodes, we prepare three types of control devices with the same WSe₂ thickness ($t_{WSe_2} \approx 9$ nm): vertical junctions with symmetric contacts of Pt–WSe₂–Pt and Gd–WSe₂–Gd, and homogeneous MoS₂ heterojunctions with an asymmetric contact of Gd–MoS₂–Pt. For the WSe₂ devices with symmetric metal contacts, as shown in Fig. 4, $I$–$V_{SD}$ curves are similar in forward and backward biases: there is no current rectification. Note that the operating currents ($I_{ON}$) of the device with Gd (Pt) contacts are higher (lower) than the ones from the asymmetric metal contact. In comparison, the vertical diodes with MoS₂ films operate with an on-current reduced by a factor of two or more, attributed to the heightened Schottky barrier at the MoS₂–Pt contact.

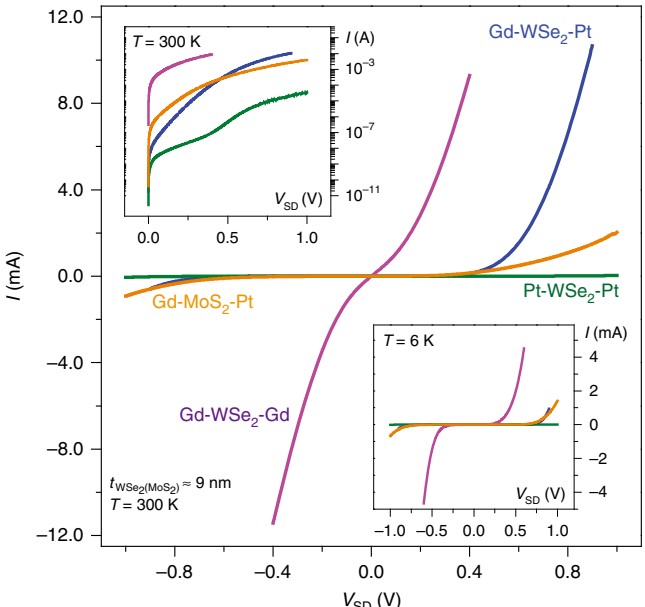

**Fig. 4** Vertical heterojunctions with symmetric vs. asymmetric metal contacts. $I$ vs. $V_{SD}$ curves at $T = 300$ and 6 K (lower inset) from the WSe₂ (blue) and MoS₂ (orange) vertical heterojunctions with asymmetric metal contacts of Pt and Gd. Vertical WSe₂ junctions with symmetric metal contacts of either Pt or Gd are plotted with green (Pt–WSe₂–Pt) and purple (Gd–WSe₂–Gd) lines. The thickness of the WSe₂ and MoS₂ films is around 9 nm ($\approx$14L). (Upper inset) $I$ vs. $V_{SD}$ characteristics at forward bias in log–log scale at $T = 300$ K, highlighting the distinctions in $I$–$V_{SD}$ curves in value among the vertical heterojunctions with different metal–2D vdW contact configurations

We extract the effective Schottky barriers formed at the contacts of Gd–WSe₂ and Pt–WSe₂ from the symmetrically contacted vertical junctions with thicker ($t_{WSe_2} > 60$ nm) films. As displayed in Supplementary Figure 10, $\Phi_{SB}$ at the Pt–WSe₂ contact ($\phi_{SB,Pt-WSe_2} \approx 530$ meV) is estimated to be three times higher than the barrier at the Gd–WSe₂ contact ($\phi_{SB,Gd-WSe_2} \approx 190$ meV) at $V_{SD} \approx 0$ V. Thus, we conjecture that the majority of currents in our Pt–WSe₂–Gd vertical diodes at elevated temperatures are from the electrons crossing over the Schottky barrier at the junction of Gd-doped and un-doped WSe₂ layers. Distinct from the symmetrically defined metal–WSe₂ junctions, however, both electrons and holes in the asymmetrically formed Pt–WSe₂–Gd devices need to overcome two barriers at the junctions of Pt–WSe₂ and WSe₂–Gd to contribute charge flows (Fig. 3b), so that the on-current level in the asymmetrically defined device becomes lower than the current in the symmetrically formed Gd–WSe₂–Gd device (Fig. 4); however, a detailed discussion on how the effective $\Phi_{SB}$ is determined in the Pt–WSe₂–Gd vertical devices is beyond the scope of current report, and it requires further in-depth theoretical and experimental attention. Additionally, we note that the asymmetric $\Phi_{SB}$ traits (Supplementary Figure 10) and $I$–$V_{SD}$ curves (Fig. 4) for forward and backward biases from the symmetrically formed junctions (Pt–WSe₂–Pt and Gd–WSe₂–Gd) are due to the asymmetric contact area in size for the top and bottom metal contacts to WSe₂ films. Bottom contacts are fixed as the area of a SiN through-hole (3 μm in diameter), but top-contact areas vary in size depending on the flakes (Fig. 1a, f)[44].

**High-frequency switching operations of WSe₂ vertical diodes.** We demonstrate the performance of our vertical WSe₂ diodes in

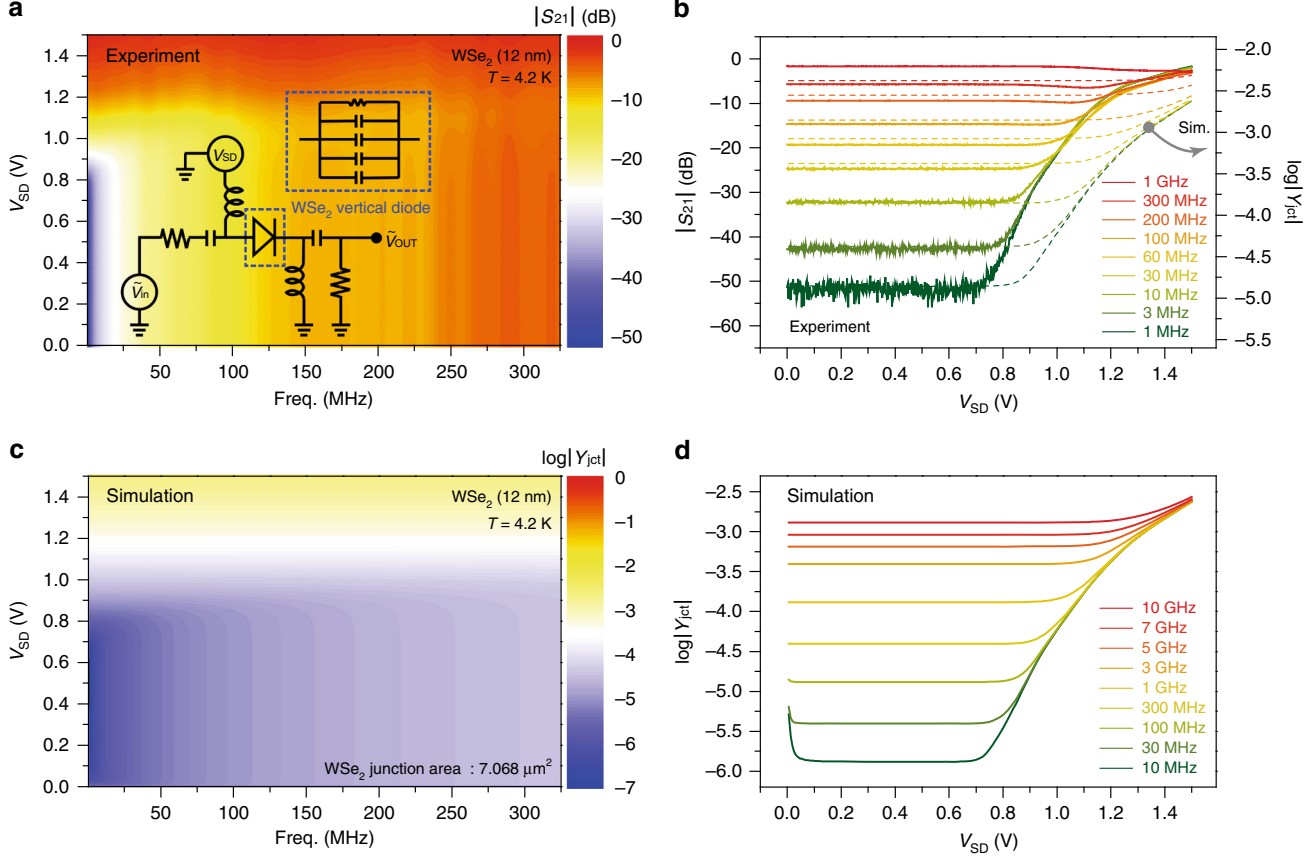

**Fig. 5** High-frequency switching operations of WSe$_2$ vertical diodes. **a** Two-dimensional display of the forward transmission coefficient ($|S_{21}|$) at varying $V_{SD}$ and a driving frequency measured at $T = 4.2$ K from the vertical diode with $t_{WSe_2} \approx 12$ nm as a p–i–n junction thickness. High-frequency switching with a threshold bias at $V_{SD} \approx 0.8$ V is observed up to $f \leq 300$ MHz. (inset) Scattering parameters are monitored through a 50-Ω network with a bias tee supplying DC currents and high-frequency inputs. Switching operations are determined by the junction resistance and the parallel connected capacitances. **b** High-frequency switching characteristics ($|S_{21}|$) at varying frequencies. The calculated junction admittances ($Y_{jct}$) are plotted as dotted lines, proving that the parasitic signal loss through the Si/SiN substrate limits the high-frequency operations in our WSe$_2$ vertical diodes. **c, d** Two-dimensional display (**c**) and line profiles (**d**) of simulated junction admittances with the sole consideration of the active WSe$_2$ vertical junction area of 7.068 μm$^2$, presenting the possibility that 2D vdW vertical diodes can be utilized in high-frequency switching operations

high-efficient switch operations in an RF domain at cryogenic temperatures, where 2D vdW-based tunneling devices are recognized to possess a superiority over conventional 3D semiconductor-based diodes[19,42,45]. Fig. 5a displays transmission coefficient ($S_{21}$) responses to a forward $V_{SD}$ at varying frequencies at $T = 4.2$ K, where the RF switching operation is observed up to 300 MHz with a threshold voltage at $V_{SD} \approx 0.8$ V from the vertical diode with a 12-nm-thick film. At $f = 1$ MHz (Fig. 5b), for example, −52 dB of isolation and a −3 dB of insertion loss can be characterized for the OFF and ON state of the switching operation. Scattering parameters are measured with a network analyzer with a bias tee, supplying DC current through the vertical diode (inset in Fig. 5a). We found out that parasitic reactive components such as high-frequency signal losses through the Si/SiN substrates (5 mm × 5 mm in size) set the frequency limit for the switching operation at $f \leq 300$ MHz in our vertical diodes. Note that the junction area where capacitive coupling is active in high-frequency setups is much larger than the actual size of the WSe$_2$ vertical junctions, defined here by a 3 μm through-hole since most of the top and bottom surfaces of the Si/SiN platform is covered with metal films (Supplementary Figure 12).

The dotted lines in Fig. 5b indicate how much admittance ($Y_{jct} = 1/Z_{jct}$) of the vertical junction, which directly relates to the transmission coefficient $|S_{21}|$, varies at different frequencies with considerations of all the capacitive couplings in the current device settings (see Supplementary Note 2). The high-frequency responses,

which agree well with experimental data, confirm that capacitive loss through the substrate is indeed the major source for limiting high-frequency operations in our WSe$_2$ vertical diodes (Supplementary Figures 13–15 and Supplementary Table 1). Figure 5c, d show how the admittance of the vertical diode with a 12-nm-thick film would respond at varying frequencies when the capacitive coupling through the active WSe$_2$ area only is accounted for (junction area = 7.068 μm$^2$). From these results, we can infer that our vertically formed 2D vdW WSe$_2$ diodes can be used for high-efficient switching operations up to $f \leq 10$ GHz and more at cryogenic temperatures.

## Discussion

In this work, we carefully analyze vertical charge transport behavior through WSe$_2$ single crystals from monolayer to multilayers in a layer-by-layer approach, and find out that optimal diode characteristics, which follow an ideal Shockley diode relation, are realized with flakes of ≈10 nm (16L) in thickness. To form a true vertical heterojunction with the homogeneous WSe$_2$ crystal, Gd and Pt films are directly evaporated on the top- and bottom-most facets of the flake, which results in an atomically well-defined p–i–n heterojunction, free of any crystallographic misalignments or defects, along with an extremely low metal–WSe$_2$ contact resistance. Fully benefitting from these features, we can achieve an ultrahigh on-current density of $J \geq 2 \times 10^5$ A/cm$^2$ at room temperature, which

could be the ultimate limit in nanoscale diode operations. Moreover, transport through our vertical diodes, governed by a series of quantum tunneling events (namely, DT, Fowler–Nordheim tunneling, and SE tunneling) is sensitive to flake thickness, which adds an additional layer number control knob in 2D vdW-based diode operations. We finally test our vertical $WSe_2$ diodes for an RF switch operation as well, with encouraging results revealing that ultrathin 2D-layered single crystals have a knack for high-efficient switching, operating up to 10 GHz and more at cryogenic temperatures. Our experimental approaches can be applicable not only to other types of vertical devices, such as ultrasensitive photodetectors and light-emitting diodes, but also to most 2D vdW materials and metals and their unlimited combinations, thus widening both experimental and theoretical research fields of 2D vdW-based device applications.

## Methods

**Device fabrication.** Our $WSe_2$ pure vertical heterojunctions are fabricated through direct metal evaporations on both facets of ultrathin homogeneous synthetic $WSe_2$ crystals, suspended on a SiN through-hole with a 3 μm diameter. The high-purity (>99.995%) $WSe_2$ and $MoS_2$ crystals were purchased from HQ Graphene with no additional dopants added during growth procedures. As illustrated in Supplementary Figures 1 and 2, we start with a high-resistivity silicon (Si) wafer with 300-nm-thick low stress SiN and 100-nm-thick $SiO_2$ films, grown on both sides of the wafer. Squared patterns of 730 μm × 730 μm in size are defined on the backside of the wafer by conventional photolithography and development process. The unmasked SiN film is etched in a reactive ion etching (RIE) system with $CF_4$ and Ar. After removing $SiO_2$ film in 6:1 diluted buffered oxide etchant (BOE) for 2 min, the 525-μm-thick silicon wafer is anisotropically etched for 10 h in 20% potassium hydroxide (KOH) aqueous solution heated to 80 °C. The remaining $SiO_2$, which works as an etch-stop layer, is further removed in BOE, and a 50 μm × 50 μm square of suspended SiN membrane is released on the Si platform. A circular-shaped through-hole is patterned by electron beam lithography using a ZEP 520A resist, and the exposed SiN layer is etched with RIE to finally form the SiN membrane with a 3 μm through-hole. Then, monolayer to multilayer $WSe_2$ flakes are mechanically exfoliated on stacks of PMMA (poly(methyl methacrylate))/PSS (poly-styrene sulfonic) layers, and transferred onto the SiN membrane by well-known dry-transfer methods. We remove the PMMA film in warm acetone for an hour and annealed the samples at 250 °C for 7 h in a mixture of Ar:$H_2$ = 9:1 to ensure residue-free suspended $WSe_2$ surfaces. Gd/Au and Pt/Au films are deposited on the top and the bottom surfaces of the suspended $WSe_2$ flake at less than a 0.3 Å/s evaporation rate in a good vacuum (≤$10^{-7}$ Torr) condition. The spatial distance between $WSe_2$ flakes and metal sources inside our e-beam evaporation chamber is more than ≈80 cm in order to minimize heat-related damage on the interfacial $WSe_2$ flakes. Finally, the bottom Si hole is filled with sputtered Ti/Au (≈20 nm/≈400 nm) films for a stable electrical connection.

## Data availability

All data supporting the findings of this study are available from the corresponding author on request.

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

## Acknowledgements

This work was supported by the research grant for the Development of Converging Measurement Technology for Nanotechnology (KRISS-2017-GP2017-0019) funded by Korea Research Institute of Standards and Science and the Korea-Hungary joint laboratory program for Nanoscience through the National Research Council of Science and Technology. This work was also supported by the Priority Research Center Program (2010-0020207), the Basic Science Research Program (NRF-2017R1D1A1B03035727), and KIST Institutional Programs and Nano Material Technology Development Program (NRF-2017M3A7B4049167) through the National Research Foundations of Korea.

## Author contributions

S.J. conceived and designed the experiments. G.N., H.K., and M.F.K. fabricated devices and performed measurements. K.S.K., D.S.L., and J.Y.H. performed TEM measurements. J.K. and J.S. set up the high-frequency measurements. G.N., H.K., D.H.S., C.H., J.E., and S.J. analyzed the data and prepared the manuscript with comments from all the authors.

## Additional information

**Competing interests:** The authors declare no competing interests.

