## [Peer Review File · Nature Communications]

Reviewers' comments:

Reviewer #1 (Remarks to the Author):

This manuscript reports a detailed study of the design and performance of vertical WSe₂ diodes. The approach suggested by authors can be used to characterize other devices of similar type. The manuscript is suitable for publication after minor revisions:

- Some font sizes (especially the legends) in Figs.2-5 should be made larger.

Reviewer #2 (Remarks to the Author):

Authors have fabricated P-I-N heterojunctions using metal-induced doping (Gadolinium and Platinum) in homogeneous WSe₂. The employment of gadolinium leads to an excellent n-type contact thanks to its extremely low work function (2.9 eV). While the work presented herein is interesting, I strongly recommend the authors to make the following modifications to the manuscript.

1) Please specify the source of WSe₂ flakes since each company/growth method has different qualities/properties.

2) Authors state that forming gas (Ar:H₂) annealing was performed to clean the polymer residues on the surface of WSe₂ flakes. The annealing temperature and treatment time is relatively high and long. Did authors check for the formation of defect/vacancy after the anneal?

3) Authors claim that heavily doped layers formed by metal contacts lead to the layer-insensitive device performances for 1~3 layers WSe₂. However, direct metal deposition processes such as evaporation can easily damage 2D materials including WSe₂, as noted by a recent publication ("Approaching the Schottky–Mott limit in van der Waals metal-semiconductor junctions" Yuan Liu et al., Nature, vol 557, p. 696). Authors should carefully examine the I-V characteristics of few-layer (less than three layers in particular) WSe₂ samples and pay close attention to the cross-sectional TEM images on the WSe₂–metals interfaces as the diffuse metal atoms can create leakage current paths.

4) For many applications, RF switching at room temperature is more pertinent than that at cryogenic temperature. Could authors provide more data at room temperature?

5) Authors explain the charge transfer through the vertical structures with the different thickness of WSe₂ flake using a series of quantum tunneling events. The change in WSe₂ band gap as a function

of the layer count could also play a non-negligible role in charge transfer. Could the authors elaborate on this?

6) Authors claim that precise doping profile engineering is helpful to optimize the device performance for electrical and optical applications. However, vertical device structure suggested in the manuscript may not be suitable for optical applications due to the metal layers at top and bottom of the 2D stack. Is there a way to alleviate the obstacle?

Reviewer #3 (Remarks to the Author):

The authors claim to have created a vertical p-i-n diode with WSe₂ sandwiched between two metals of dissimilar work functions. Using low workfunction Gd and high workfunction Pt, the authors have tried to demonstrate the creation of a p-n junction, which can be modulated into a p-i-n junction with controllable thickness of the intrinsic WSe₂ using the thickness of WSe₂ as the knob.

1. The main concern in the diode I-V demonstration that I have is how much of this current is due to the tunneling from one metal layer to the other through the WSe₂ barrier. Several reports demonstrate tunneling through TMDCs or hBN when metal or graphene are used as electrodes (Britnell et al., Science, vol. 335, 2012; and several other reports from the Novoselov group between 2012 and 2014). Surely a part of the current that the authors show as diode I-V is due to this tunneling component. The symmetric I-V till WSe₂ thickness of 5 layers is a strong indication of tunneling from one metallic electrode to the other through WSe₂. For higher thicknesses of WSe₂, there could be the metal-workfunction-induced doping effects, but that essentially brings an asymmetry in the WSe₂ barrier itself. Thus, the tunneling current looks asymmetric. I feel that much of the current observed is not due to the p-i-n junction effect, but rather is due to the asymmetric barrier effect leading to asymmetry in tunneling current.
2. The p-i-n diode current through WSe₂ will be a result of the density of states in this material. Can the authors show some density of states calculations to prove that the observed current is indeed from the WSe₂ p-i-n diode formation and not simply due to tunneling through the WSe₂ barrier?
3. The authors observe linear I-V for 1-3 L of WSe₂. However a recent report (Ruijing Ge, et al., Nano Letters, 2018, 18, 434-441) show the tunneling characteristics with monolayer MoS₂, WS₂, WSe₂ and MoSe₂ sandwiched between gold electrodes as well as between gold and graphene electrodes. Can the authors comment on this difference in their observed I-V?
4. What do the authors mean by “metallic junctions”? This is not a standard terminology in device physics.

5. The rectification observed for ≤ 16 layers of WSe₂ is rather low. A p-n junction should definitely be able to provide larger rectification ratios. Can the authors comment on this?
6. The highest rectification ratio observed is for layer thickness $> 50 L$. The thickness of WSe₂ barrier here would be ~ 35 nm. Certainly this barrier is too thick for tunneling from one metal electrode to another. It is highly possible that the p-i-n diode is getting formed under these circumstances and the effect of the diode is dominating in the I-V. But, the thickness of WSe₂ used does not correspond to a 2D material.
7. Several typos. Example: Figure 5a "Experimnet", page 7: "Schokley",...

Reviewer(s)' Remarks to the Author:

Reviewer #1

Comments:

This manuscript reports a detailed study of the design and performance of vertical WSe₂ diodes. The approach suggested by authors can be used to characterize other devices of similar type. The manuscript is suitable for publication after minor revisions: Some font sizes (especially the legends) in Figs.2–5 should be made larger.

Response:

We thank the Reviewer for their high praise on the quality of our experiments and for their recommendation for publication in *Nature Communications*. As the Reviewer suggested, we have enlarged the fonts in Figs. 1–5, and the newly updated figures surely make the viewgraphs represent our data more clearly.

Reviewer #2

Comments:

Authors have fabricated P-I-N heterojunctions using metal-induced doping (Gadolinium and Platinum) in homogeneous WSe₂. The employment of gadolinium leads to an excellent n-type contact thanks to its extremely low work function (2.9 eV). While the work presented herein is interesting, I strongly recommend the authors to make the following modifications to the manuscript.

Response:

We thank for the Reviewer for finding our work interesting, and for the thoughtful comments and questions to make our manuscript better. Detailed point-by-point responses to each question are listed below, and all alterations to the manuscript are highlighted here in red. We hope our responses/revisions fully resolve the concerns that the Reviewer has raised.

1) Please specify the source of WSe₂ flakes since each company/growth method has different qualities/properties.

Response:

We regret that this information wasn't included in the Method section (Device Fabrication) in the originally submitted manuscript. We purchased WSe₂ and MoS₂ single crystals from HQ

Graphene (www.hqgraphene.com), with the provider's guarantee that the crystals, synthesized by chemical vapor transport method, are of high purity (> 99.995%) with no additional dopants added during growth. We have added such information in the Method section as follows.

Added in the main text (Methods Section):

Device Fabrication. Our WSe₂ pure vertical heterojunctions are fabricated through direct metal evaporations on both facets of ultrathin homogeneous **synthetic** WSe₂ crystals, suspended on a silicon nitride (SiN) through-hole with a 3 μm diameter. **The high-purity (> 99.995%) WSe₂ and MoS₂ crystals were purchased from HQ Graphene with no additional dopants added during growth procedures.**

2) Authors state that forming gas (Ar:H₂) annealing was performed to clean the polymer residues on the surface of WSe₂ flakes. The annealing temperature and treatment time is relatively high and long. Did authors check for the formation of defect/vacancy after the anneal?

Response:

We appreciate the Reviewer's genuine concerns on our sample preparation. Indeed, we devoted serious efforts to properly optimize the thermal treatment procedures with transition metal dichalcogenide (TMD) films without generating post-annealed defects, vacancies, or structural deformations. It was previously confirmed by others that high temperature annealing at $T \approx 900$ K under ultrahigh vacuum conditions actually produces atomic-defect states in the monolayer MoS₂ film, although no visible defect states are formed at a moderate temperature of $T \approx 400$ K [M. Liu *et al.* Temperature-Triggered Sulfur Vacancy Evolution in Monolayer MoS₂/Graphene Heterostructures, *Small* **13**, 1602967 (2017)]. Although we used a significantly lowered annealing temperature of $T = 250$ °C, when compared with the temperature for treating graphene films at $T = 350$ °C, we still needed to independently confirm whether defect-related states would be generated in the TMDs after our thermal treatment procedure at 250 °C.

The post-annealed TMD film quality can be examined by photoluminescence (PL) and electron tunneling spectroscopy. In general, PL spectra related to the defect states appear as peaks around ≈ 775 nm and ≈ 920 nm, and the defect-induced PL peaks are strongly enhanced as the number of defects increases [Z. Wu *et al.* Defect Activated Photoluminescence in WSe₂ Monolayer, *J. Phy. Chem. C.* **121**(22), 12294 (2017)]. The PL spectrum, shown below in Fig. R1a, reveals two representative PL peaks at around 1.67 eV (741 nm) and 1.71 eV (724 nm), and these two peaks respectively represent the recombination of trions and neutral excitons of the monolayer WSe₂ film at $T = 80$ K. The monolayer WSe₂ film was annealed at 250 °C before PL measurements. The sharpness of the neutral excitonic peak, represented by the full-width-at-half-maximum (FWHM ≈ 3.76 nm) and the absence of defect-related PL peaks, confirms that monolayer WSe₂ films are intact without suffering from post-annealed defects during our high-temperature ($T = 250$ °C) annealing procedures. In contrast, the PL spectrum taken from a monolayer WSe₂ film after 350 °C annealing reveals a glimpse of defect-related spectra features, as displayed in Fig. R1b. Although the FWHM of the neutral excitonic peak is still quite narrow

(≈ 4.4 nm) even after 350 °C annealing, we chose the thermal treatment temperature for TMDs to be 250 °C.

Figure R1: Photoluminescence spectra from the monolayer WSe₂ film after (a) 250 °C and (b) 350 °C thermal annealing treatments.

In addition, we are able to investigate the electronic structures of monolayer WSe₂ films with electron tunneling spectroscopy measurements, and the following key material parameters (such as the relative positions of the conduction and valence bands with respect to the Fermi level of WSe₂) are accurately determined. While preparing tunneling devices, which require multiple steps of 2D-layer transfers, we need to repeatedly anneal the monolayer WSe₂ film at 250 °C not less than four to five times. Tunneling spectra, which we will present in a separate manuscript soon, indicate that the Fermi level (E_F) of the monolayer WSe₂ film still remains in the middle of the energy gap with E_F being slightly closer to the valence-band edge, confirming that monolayer WSe₂ is a weakly doped p-type semiconductor, as consistent with broadly accepted intrinsic WSe₂ films. Therefore, based on optical (PL) and electron tunneling spectroscopy studies, we are assured that our thermal annealing procedures at $T = 250$ °C in the mixture of Ar:H₂ do not generate any defect states or defect-related dopants.

3) Authors claim that heavily doped layers formed by metal contacts lead to the layer-insensitive device performances for 1~3 layers WSe₂. However, direct metal deposition processes such as evaporation can easily damage 2D materials including WSe₂, as noted by a recent publication (“Approaching the Schottky–Mott limit in van der Waals metal-semiconductor junctions” Yuan Liu et al., Nature, vol 557, p. 696). Authors should carefully examine the I-V characteristics of few-layer (less than three layers in particular) WSe₂ samples and pay close attention to the

cross-sectional TEM images on the WSe₂–metals interfaces as the diffuse metal atoms can create leakage current paths.

Response:

We thank the Reviewer for their valuable suggestions on the careful investigation of vertical charge transport through mono- to few-layer WSe₂ flakes. Indeed, we have been aware of the possible metal-atom diffusions to thin flakes and their implications to vertical charge flows, as suggested in the recent publication by Yuan Liu *et al.* in Nature, **557**, p. 696. The TEM image presented in this paper suggests that Au atoms can diffuse through the interfacial MoS₂ up to several layers and possibly form Au-atom filaments through semiconducting channels, such that vertical charge flows could be dominated by so-called leakage current through the metallic junctions.

In our WSe₂ vertical junctions, however, we have not observed any charge transport characteristics that could be related to diffused metal atoms through semiconducting channels. In general, metallic and semiconducting junctions can be identified by temperature-dependent *I-V* characteristics: the resistance of metallic junctions increases in value as the temperature increases, while semiconducting junctions show much higher resistances at lower temperatures. Although the *I-V* characteristics from the monolayer WSe₂ vertical junction, as shown below in Fig. R2a, reveal a slight increase in junction resistance when temperature increases from 6 K to 300 K, the resistance difference is too small to be regarded as a metallic junction. Moreover, junction resistances for the bi-layer (Fig. R2b) and the tri-layer (Fig. R2c) WSe₂ flakes at $T = 6$ K are even smaller than the resistances at $T = 300$ K, which is not related to ‘could be formed’ metallic channels from diffused metal atoms. In general, resistances from metallic channels, even in a form of nano filament, show a strong positive temperature dependence, as demonstrated in the following reports. [(1) Z. Cheng *et al.* Temperature Dependence of Electrical and Thermal Conduction in Single Silver Nanowire, *Sci. Rep.* **5**, 10415 (2015), (2) H. Rho *et al.* Metal nanofibrils embedded in long free-standing carbon nanotube fibers with a high critical current density, *NPG Asia Mater.* **10**, 146 (2018) (3) A. Bid *et al.* Temperature dependence of the resistance of metallic nanowires of diameter >15 nm: applicability of Bloch-Gruneisen theorem, *Phys. Rev. B*, **74**, 035426 (2006).] Additionally, we have checked five WSe₂ junctions with TEM, and no conclusive evidence for metal-atom induced defects or diffusion through the WSe₂ channels was identified.

Here, we do not claim that our vertical junctions could be immune to any defects or damage from Gd and Pt metal evaporations. Rather, we believe that the confined area of interest for our WSe₂ vertical junctions—a 3 μm diameter circle—can be the main attribute for the absence of experimental signatures of interfacial defects. This is further helped by our nonviolent Gd and Pt metal evaporations, with an extremely small evaporation rate of less than 0.3 $\text{\AA}/\text{s}$, and plus, the distance between WSe₂ flakes and metal sources (more than ≈ 80 cm) is thought to minimize the so-called heat-related damage to the flakes. We have added the following explanations on the possibility of metal-induced deformation in WSe₂ films in the main text with a new reference cited.

Figure R2: I - V characteristic curves for the WSe_2 vertical junctions of (a) mono-, (b) bi-, (c) tri-, and (d) four-layer flakes measured at $T = 300$ K (red) and $T = 6$ K (blue).

Added in the main text (pages 7–8):

Note that the I - V_{SD} curves at $T = 300$ K from the devices with one-, two- and three-layer flakes are indistinguishable, mainly due to the fact that the junction resistances of these heavily doped layers become comparable to or even smaller than the combined contact resistance ($\leq 15 \Omega$) of the Gd- WSe_2 and WSe_2 -Pt junctions. **We point out that here, in our vertical junctions, we cannot find any charge transport characteristics that could be related to plausible metal-induced structural defects or metal-atom diffusions through the semiconducting films [Cite, Yuan et al. Nature, 557, 696], mostly due to the confined area of interest of our WSe_2 vertical junctions ($\approx 3 \mu\text{m}$ in diameter) and the carefully controlled Pt and Gd metal evaporation conditions.**

Added in the main text (Method session):

Device Fabrication. Gd/Au and Pt/Au films are deposited on the top and the bottom surfaces of the suspended WSe_2 flake at less than a $0.3 \text{ \AA}/\text{sec}$ evaporation rate in a good vacuum ($\leq 10^{-7}$ Torr) condition. **The spatial distance between WSe_2 flakes and metal sources inside our e-beam evaporation chamber is more than ≈ 80 cm in order to minimize heat-related damage on the interfacial WSe_2 flakes.** Finally, the bottom Si hole is filled with sputtered Ti/Au ($\approx 20 \text{ nm}/\approx 400 \text{ nm}$) films for a stable electrical connection.

4) For many applications, RF switching at room temperature is more pertinent than that at cryogenic temperature. Could authors provide more data at room temperature?

Response:

We thank the Reviewer for reminding us which operating conditions are more pertinent for practical RF switching applications. Although we focused on switching characteristics at low temperatures to specify the roles of the WSe₂ vertical junctions in high-frequency domains, our vertical junctions also reveal reasonable RF switching characteristics at room temperature, as shown in Fig. R3 below. Per the Reviewer's request, we have added high-frequency switching operation data measured at room temperature in the Supplementary Information.

Added in the Supplementary Information (Section S5)

Updated Figure S16 and caption.

Figure S16. (a) Transmission coefficient responses to a forward V_{SD} by varying frequencies at $T = 300$ K, where the RF switching property is observed up to $f = 200$ MHz. The transmission coefficient $|S_{21}|$ varies at different frequency regimes. (b) At $f = 1$ MHz, -43 dB of isolation and -5 dB of insertion loss can be characterized for the OFF and ON state of the switching operation. However, the difference between the ON and OFF states gradually reduces with increasing frequencies. (c) Simulation result of AC admittance variations up to $f \leq 1$ GHz at $T = 300$ K considering only the capacitive coupling through the active WSe₂ area (junction area = 7.068 μm^2).

Figure R3: High-frequency switching operation of the WSe₂ vertical diode ($t_{\text{WSe}_2} \approx 12$ nm) at $T = 300$ K.

5) Authors explain the charge transfer through the vertical structures with the different thickness of WSe₂ flake using a series of quantum tunneling events. The change in WSe₂ band gap as a function of the layer count could also play a non-negligible role in charge transfer. Could the authors elaborate on this?

Response:

We appreciate these reasonable concerns on the detailed mechanisms for vertical charge flows influenced by varying energy bands at different layer thicknesses. Indeed, the layer dependency

of the (quasiparticle or optical) energy band gap of WSe₂ and other TMDs has been widely studied and well understood. For example, the optical energy gaps of WSe₂ films decrease its size from ≈ 1.7 eV in monolayer [Yun *et al.* Physical Review B **85** 033305 (2012)] to an indirect energy gap of ≈ 1.0 eV in bulk forms [Prakash *et al.*, ACS Nano, **11**, 1626 (2017)]. Although the tunability of the energy gaps of WSe₂ films at varying thickness—especially for few-layer WSe₂ films—could induce a non-negligible impact on vertical charge transport, the majority of vertical charge flows are driven by direct tunneling through the junctions made of six WSe₂ layers and less, at which the implications of energy-gap variations on the vertical charge flows would be minimal when compared with the Schottky tunneling events for thicker films. According to a previous report [Prakash *et al.*, ACS Nano, **11**, 1626 (2017)], moreover, the size of WSe₂ energy band gaps becomes saturated at around ≈ 1 eV for WSe₂ flakes with 10 layers and more, so that our vertical WSe₂ junctions ($t_{\text{WSe}_2} \geq 10$ L), where ultimate performance of WSe₂ vertical diodes has been realized, would not be influenced by the energy-gap variations of the WSe₂ films.

As presented in the Supplementary Information (Fig. S9), our studies independently confirm that Schottky tunneling events start dominating vertical charge flows in WSe₂ junctions with 10 layers and more, while the quantum tunneling events dominating the vertical junctions with less than 10 WSe₂ layers are direct (DT) and Fowler–Nordheim (FN) tunneling in which Schottky barrier heights or energy band gaps are less important.

6) Authors claim that precise doping profile engineering is helpful to optimize the device performance for electrical and optical applications. However, vertical device structure suggested in the manuscript may not be suitable for optical applications due to the metal layers at top and bottom of the 2D stack. Is there a way to alleviate the obstacle?

Response:

We thank the Reviewer for this concern about the optical applications of our vertical device platform. In the current device outline, we deposited ≈ 20 nm of Gd and ≈ 10 nm of Pt respectively for the top and bottom metal–WSe₂ contacts, and an additional ≈ 70 nm of Au was deposited on top of Gd and Pt layers to form protection layers, preventing oxidization especially for the Gd films. Indeed, a large portion of incident light photons are absorbed in the Au films before transmission through the films to be absorbed by photoactive media [Axelevitch *et al.* in Physics Procedia **32**, 1 (2012)] (Figure R4). Thus, it is expected that a meager 10% or less portion of light, which is still good enough for some photodetectors, can be absorbed by WSe₂ in the current device structure, requiring us to search for better device configurations to enhance its optoelectronic responses.

Actually, for follow-up studies on the current manuscript, we have dedicated our research efforts to relieve the above-mentioned issue, and some positive data proves that our approach can be the right one. In short, we reduced Gd and Pt film thicknesses to ≈ 5 nm and used a minimal thickness of Au protection layer, like ≈ 10 nm or less. Pt film is reversely deposited on the top facet of TMD films, where incident photons are shined, which allows us to significantly reduce the thickness of the Au-protection layers thanks to the better stability of Pt films when compared with Gd. With this revised device scheme, *I-V* characteristics are measured that do not differ

from devices with thicker metal films; we hope the complete optoelectronic responses to be presented in upcoming manuscripts soon.

Besides reducing the metal film thickness, we can also implement graphene as a top contact material, although the vdW gap between the graphene and TMD layers can add an additional resistance channel. Otherwise, conventional transparent conducting oxides such as indium tin oxide (ITO) or indium zinc oxide (IZO) could be a wonderful alternative for a top contact electrode as the well-known work-function tunability of those conducting films can add additional exciting experimental aspects for the optoelectronic applications of 2D vertical structures.

Figure R4: Transmittance of gold films deposited by thermal evaporation (A. Axelevitch *et al.* Physics Procedia **32**, 1 (2012)).

Reviewer #3

Comments:

The authors claim to have created a vertical p-i-n diode with WSe₂ sandwiched between two metals of dissimilar work functions. Using low workfunction Gd and high workfunction Pt, the authors have tried to demonstrate the creation of a p-n junction, which can be modulated into a p-i-n junction with controllable thickness of the intrinsic WSe₂ using the thickness of WSe₂ as the knob.

1. The main concern in the diode I-V demonstration that I have is how much of this current is due to the tunneling from one metal layer to the other through the WSe₂ barrier. Several reports demonstrate tunneling through TMDCs or hBN when metal or graphene are used as electrodes (Britnell et al., Science, vol. 335, 2012; and several other reports from the Novoselov group between 2012 and 2014). Surely a part of the current that the authors show as diode I-V is due to this tunneling component. The symmetric I-V till WSe₂ thickness of 5 layers is a strong indication of tunneling from one metallic electrode to the other through WSe₂. For higher thicknesses of WSe₂, there could be the metal-workfunction-induced doping effects, but that essentially brings an asymmetry in the WSe₂ barrier itself. Thus, the tunneling current looks asymmetric. I feel that much of the current observed is not due to the p-i-n junction effect, but rather is due to the asymmetric barrier effect leading to asymmetry in tunneling current.

Response:

We thank for the Reviewer for their efforts to read our manuscript thoroughly, and for raising genuine concerns on the detailed transport mechanisms of vertical charge flows through our WSe₂ p-i-n heterojunctions. In a word: yes! The diode I-V characteristics in our vertical junctions, especially those made of WSe₂ flakes with 10 layers or more, are indeed governed by Schottky emission tunneling events where asymmetric Schottky barriers play an important role in diode operations, as the Reviewer suggests, and as we stressed in the manuscript. The vertical charge flows through the devices with less than 10 WSe₂ layers are sequentially determined by direct tunneling (DT) and Fowler–Nordheim (FN) tunneling events as the layer number increases from mono- to several layers.

Here, we need to assert that our vertical WSe₂ p-i-n heterojunctions and their transport characteristics are quite unique from conventional bulk semiconducting p-n diodes. In regular p-n semiconducting diodes, the forward and backward diode operations are dictated by the varying thickness of the depletion region at the p-n interfaces. In our WSe₂ heterojunctions and any other heterostructures of two-dimensional (2D) van der Waals (vdW) materials, however, the depletion region does not exist at the interfaces, mainly due to the physical spacing between layers, which is called the vdW gap. As graphically illustrated in Fig. 1b, and Figs. 3a–3c, the top- and bottom-most layers of the WSe₂ flakes become strongly hybridized with evaporated Gd and Pt films, and the Fermi level of the WSe₂ layer that is directly contacted by metal films is significantly shifted toward either the conduction (Gd–WSe₂) or the valence (Pt–WSe₂) bands due to metal-induced doping effects. Because of the vdW gap between WSe₂ layers, however, the work function-induced doping effects become significantly attenuated at the second layer, and more at the third

layer, while the next layers remain more or less intrinsic, which we assign as the *i*-WSe₂ region. The insulating *i*-WSe₂ region is weakly influenced by the forward or backward direction of driving currents; this non-varying insulating area makes our WSe₂ vertical *p-i-n* heterojunctions or any other 2D vdW-based heterostructures distinct from conventional *p-n* diodes. Because of this particular physical configuration, our vertical WSe₂ diodes possess the unique device characteristics that both conventional 3D semiconductor-based Schottky diodes and regular *p-i-n* diodes can provide, as we stress in the current manuscript.

Following this comment, we admit that we failed to elaborate the difference between 2D vdW-based *p-n* heterojunctions and conventional bulk semiconductor *p-n* junctions, as to the physical formation of depletion or insulating regions at semiconductor interfaces. Thus, to clarify this important issue for readers, we have added the following sentence in the revised version of the main manuscript.

Added in the main text (page 6):

Within our operation voltage range ($|V_{SD}| \leq 1$ V) in the vertical diodes, the intrinsic WSe₂ can be considered as an insulating layer. **Here we note that the intrinsic WSe₂ layers in our vertical diodes are quite unique when compared with the insulating area defined by the depletion region in conventional bulk semiconducting *p-n* heterojunctions. While the width of the depletion region (i.e., the size of the insulating region) is tuned by the polarity and magnitude of V_{SD} in regular *p-n* diodes, the *i*-WSe₂ layers are less dependent on varying V_{SD} thanks to weaker interlayer interactions across the vdW gap in 2D layered vdW materials.** Figure 1c shows a transmission electron microscopy (TEM) image displaying...

*2. The *p-i-n* diode current through WSe₂ will be a result of the density of states in this material. Can the authors show some density of states calculations to prove that the observed current is indeed from the WSe₂ *p-i-n* diode formation and not simply due to tunneling through the WSe₂ barrier?*

Response:

As we elaborated above in our response to the first question, our vertical WSe₂ *p-i-n* heterojunctions differ from conventional bulk semiconducting *p-i-n* diodes, and charge flows through the vertical junctions are indeed governed by sequential quantum tunneling events: DT for the junctions with mono- to six-layer, FN for the junctions up to 13 L, and Schottky emission tunneling for thicker WSe₂ films. Below in Fig. R1, we schematically illustrate how energy band alignments are formed as the layer number increases. Note that the metal-induced doping areas, respectively denoted as *p*-WSe₂ and *n*-WSe₂ are fixed in width while the width of insulating region, *i*-WSe₂ increases as the film becomes thick. Fig. R1a indicates the regime where DT events prevail through thin tunnel barrier, and FN tunneling becomes important for the films up to 13 L as in Fig. R1b. When films become much thicker (Figs. R1c and R1d), effects from both DT and FN tunneling are subsidized with Schottky emission (SE) across the Schottky barrier heights become a dominant transport mechanism across the vertical WSe₂ junctions. In a

backward V_{SD} , the Schottky barrier heights for SE tunneling are much higher than those for forward V_{SD} , resulting in an efficient current rectification for thick WSe_2 junctions, ≥ 13 L.

Figure R1: Energy band diagrams of the vertical WSe_2 p - i - n heterojunctions at a forward V_{SD} as the layer number increases.

3. The authors observe linear I - V for 1-3 L of WSe_2 . However a recent report (Ruijing Ge, *et al.*, *Nano Letters*, 2018, 18, 434-441) show the tunneling characteristics with monolayer MoS_2 , WS_2 , WSe_2 and $MoSe_2$ sandwiched between gold electrodes as well as between gold and graphene electrodes. Can the authors comment on this difference in their observed I - V ?

Response:

We appreciate the Reviewer's comment and suggestion about this article demonstrating nice electronic applications with a 2D TMD-based vertical device that we should have cited in the original manuscript. We have updated our reference list to include the article by R. Ge *et al.* [*Nano Letters*, 2018, **18**, 434-441] in the revised version.

In our WSe_2 vertical junctions, we observe ohmic behaviors in I - V characteristic curves at room temperature for mono-, bi- and tri-layer WSe_2 vertical junctions, which is distinct from the resistive switching properties reported by Ruijing Ge *et al.* Although ohmic I - V characteristics were also observed in their devices, especially in the low-resistance state, we need to stress that there exists structural difference between our vertical diodes and their resistive switching memory devices. The resistive switching devices reported by R. Ge *et al.* have atomically sharp and clean metal-TMD interfaces with Au or graphene as metal contacts. As we have pointed out previously, a physical vdW gap is formed at the graphene-TMD interface that works as a tunneling barrier for vertical charge flows. Moreover, Au-TMD contacts are reported to possess a vdW gap as well, unlike strongly hybridized metal-TMD contacts such as the Gd- WSe_2 and Pt- WSe_2 junctions in our vertical structures. [A. Allain *et al.* *Nat. Mater.* **14**, 1195-1205 (2015)] Thus, the presence of vdW gaps at the metal-TMDs contacts in their devices causes the I - V characteristics to differ from our observations, especially for mono- and bi-layer junctions. Note that, in our device made of a three-layer WSe_2 flake, non-ohmic behavior can be identified at $T = 6$ K in the I - V curve below (Fig. R2c), which can be attributed to the vdW gap existing at the middle of the junction.

Figure R2: I - V characteristic curves for the WSe₂ vertical junctions of (a) mono-, (b) bi-, (c) tri-, and (d) four-layer flakes measured at $T = 300$ K (red) and $T = 6$ K (blue).

4. What do the authors mean by “metallic junctions”? This is not a standard terminology in device physics.

Response:

We deeply appreciate this correction of our misused terminology. We have replaced “metallic junctions” with “ohmic junctions” in the revised manuscript.

5. The rectification observed for ≤ 16 layers of WSe₂ is rather low. A p - n junction should definitely be able to provide larger rectification ratios. Can the authors comment on this?

Response:

We appreciate the Reviewer’s comment. As we have emphasized, our vertical p - i - n heterojunctions and any other 2D vdW-based p - n heterostructures are free of the depletion region by nature, and this absence of the depletion region is the key ingredient that makes 2D vdW-based devices unique from conventional bulk semiconducting devices. In conventional bulk diodes, the depletion region is expanded (shortened) in width when a backward (forward) V_{SD} is applied, which could result in superior rectification characteristics. In our WSe₂ vertical junctions, current rectifications originate from the asymmetric Schottky barrier heights for FN and mostly Schottky emission tunneling when a forward and reverse V_{SD} applied, as we schematically depicted in Figs. 3b (forward V_{SD}) and 3c (backward V_{SD}). Thus, for the devices with 16 or more WSe₂ layers, where Schottky emission is dominant for vertical charge flows, the rectification ratio (RR) becomes enhanced (Fig. 2c). In contrast, rectification behaviors are

gradually moderated as FN and DT tunneling effects become dominant for the devices with ≤ 16 L.

We agree that the RR from the device with 16 L is not as large as what conventional *p-n* diodes can provide, although $RR \approx 100$ is on par with other 2D vdW-based diode characteristics reported from other groups: [Cheng, R. *et al.* Nano Lett. **14**, 5590 (2014)], [R. Zhou *et al.* Nano Lett. **17**, 4787 (2017)], and [Li. *et al.* Nat. Comm. **6**, 6564 (2015)]. Despite this, we should note that our vertical heterojunction made of 16 L WSe₂ flake reveals ideal diode characteristics with low turn-on voltage and extremely high on-current density, with a reasonable RR that is still comparable to other 2D diodes. Additionally, as our data suggests, device applications requiring better RR can be simply achieved by increasing WSe₂ thickness to relatively moderate on-current density.

6. The highest rectification ratio observed is for layer thickness > 50 L. The thickness of WSe₂ barrier here would be ~ 35 nm. Certainly this barrier is too thick for tunneling from one metal electrode to another. It is highly possible that the p-i-n diode is getting formed under these circumstances and the effect of the diode is dominating in the I-V. But, the thickness of WSe₂ used does not correspond to a 2D material.

Response:

We appreciate this concern. Indeed, the comment on ≥ 35 nm WSe₂ films being too thick for tunnel barriers makes our WSe₂ vertical heterojunctions made of those thick films ideal for diode characteristics as far as higher RR is first-and-foremost concerned. As we stated in the manuscript and depicted in the energy-band diagrams (Figs. 3b and 3c), the majority of backward current is determined by DT and FN tunneling, which become significantly diminished for thicker films. In contrast, the forward charge carriers initiated by Schottky emission tunneling are able to transport at either conduction or valence bands, and those band transports become limited by space-charge effects or other scattering events for much thicker films. Our measurements suggest that, in a device with ≈ 64 L where RR is the highest at $RR \geq 10,000$, the backward current is significantly suppressed due to the thick WSe₂ film but the forward current is not yet much limited by space-charging effects or other scattering events.

7. Several typos. Example: Figure 5a “Experimnet”, page 7: “Schokley”,...

Response:

We greatly appreciate the Reviewer’s efforts and thank them for the corrections. We have carefully revised the manuscript and corrected all the typos.

REVIEWERS' COMMENTS:

Reviewer #2 (Remarks to the Author):

The authors have revised the manuscript to my satisfaction. I also note that great effort has been made to appease the other reviewer comments. The revised manuscript is suitable for publication in Nature Communications.

Reviewer #3 (Remarks to the Author):

My questions were answered satisfactorily.